# Influence of LWE on Strength of Welded Joints of HSS S960—Experimental and Numerical Analysis

**DOI:** 10.3390/ma13030747

**Published:** 2020-02-06

**Authors:** Ihor Dzioba, Tadeusz Pala

**Affiliations:** Department of Machine Design, Faculty of Mechatronics and Mechanical Engineering, Kielce University of Technology, Al. 1000-lecia PP 7, 25-314 Kielce, Poland; tpala@tu.kielce.pl

**Keywords:** HSS S960, welding, strength properties, fracture toughness, FEM modeling

## Abstract

This paper presents a strength analysis of joints made during high-strength steel S960 welding. Joints obtained by conventional and laser welding were tested. The most attention was focused on assessing the strength of the material at Heat Affect Zone (HAZ). To this aim, the effect of Linear Welding Energy (LWE) on changes in microstructure and material characteristics was studied. Numerical models of welded joints were developed using the FEM ABAQUS program. The modelled joints were subjected to simulation loads, which allowed to determine areas (the weakest links) of joints in which the destruction process may develop. Good compatibility of the strains fields on the outer surfaces of the joints calculated numerically and recorded by means of the GOM video system was obtained. Based on the tests carried out, it can be concluded that the use of welding with low levels of LEW allow obtaining joints with comparable strength to the base material.

## 1. Introduction 

In recent years, there has been an increase in interest in high-strength steels produced by thermomechanical machining methods on production lines, known as Advanced High Strength Steel (AHSS). This is due to the advantages they have—high yield strength and tensile strength with high relative elongation, of up to 20%–25% [1,2,3,4,5,6,7,8,9,10]. Steels of this type are also intended for plastic forming and machining. High-strength steels are mainly used where it is important to reduce the weight of a given structure while maintaining or increasing its load capacity. Hence, this steels is often used in the automotive industry, for the production of crane equipment, machine booms, cargo-handling equipment for maritime transport, construction machinery, railcar frames, mobile military bridges, drilling platforms, pressure tanks, oil or gas transport pipes, and in the shipbuilding industry used in environmental conditions in the temperature range −60–20 °C [11,12,13,14]. 

Most of the steel elements currently produced are joined by welding methods. The right choice of welding method and parameters is important so that the joints made are of high quality and the welding process ensures adequate performance. The highest quality of the welded joint is obtained when the properties of the weld and heat affected zones are close (or slightly higher) to the properties of the base material. Much attention is currently being given to the welding of AHSS [15,16,17,18]. Despite significant progress in the production of high-strength steels and improvement of their weldability, there are still difficulties in obtaining the strength of the welded joint at the level of strength of the base material. When welding high-strength steels, the level of used welding energy has a significant impact on the strength of the welded joints. This is due to the impact of the heat source in the welding process on the base material which is high-strength steel. Too-fast cooling of the joint favors the creation of hardening structures often with reduced impact strength, while the introduction of too much heat causes the creation of a tempered zone with reduced strength and hardness [19,20]. Microstructural changes that occur during welding also affect AHSS crack resistance [19,20,21,22]. Therefore, new welding methods using lasers and laser-conventional (hybrids) were developed for joints of AHSS [22,23,24,25,26]. These are new welding methods that require comprehensive testing to be widely used in industry. For AHSS, the issue of the level of heat consumed in the welding process becomes very important, because the joint mechanical characteristics, strength, and crack resistance depend on it. A number of publications relate to modeling of the welding process and shaping mechanical characteristics in the weld material and HAZ. There are also scientific publications on the analysis of the integrity of welded joints using their modeling and load simulation using numerical finite elements methods (FEM) [27,28,29].

The paper presents the results of tests carried out on welded joints of high-strength S960 steel plates. Various technologies were used during welding—conventional and laser welding. Welding regimes of the analyzed joints also differed in the heat consumed, assessed by the linear welding energy (LWE). As a result of experimental tests, strength and fracture toughness characteristics of metal from various zones of welded joints were determined. Based on these data, numerical models were created, and a load simulation of welded joints made according to different technologies and with different level of LWE was carried out. Numerical simulation of joints loading by tension allowed to determine areas where the highest levels of deformation occur, and where their destruction (breaking) can potentially take place. The results of numerical simulations were verified during loading of specimens containing welds in the tensile test. While loading, strain fields on surface the specimens were recorded using the video system GOM–Aramis with digital image correlation (DIC), and recorded results was compared to strains obtained on the basis of numerical calculations. Also, microstructure tests of S960 steel were carried out in various zones of welded joints.

## 2. Base Material—Specification of Welding Technology

The experimental tests were conducted on welded joints made of an advanced low-carbon and low-alloy ultra-high-strength steel S960. The mechanical properties of steel according to EN 10025-6 were: *R*_p0.2_ ≥ 960 (MPa); *R*_m_ ≥ 1000 (MPa); *A*_5_ ≥ 10 (%); CVN ≥ 27 (J) (at *T* = −40 °C); HV10 ≥ 340. Based on metallographic observation and hardness measurements, the microstructure of S960 steel was defined as a mixture of tempered martensite and bainite with grain size about 5–15 μm, which was obtained as result of thermo-mechanical treatment applied during manufacturing (Figure 1). Bainite and tempered martensite are distinguished based on the orientation of the carbide precipitates. Lower bainite has small carbides particles that are parallel to each other. Upper bainite has larger carbides, which are between the lath bainitic ferrite. The tempered martensite has carbides, which are orientated along crystallographic planes (not parallel). 

***Welding technologies***. The joined materials were AHSS S960 panels with a thickness of 6 mm and 8 mm, whose surface was properly prepared by milling and cleaning. Basic welding procedures and parameters are given in Table 1. Welded joints were made with two methods: Conventional gas metal arc welding (GMAW) and laser welding method. GMAW welded joints were made with two pass, with the temperature between passes being 150 °C and preheating to 100 °C was also used. During welding, filler materials with different strengths were used (Table 1). The shielding gas was a two-component M21 gas mixture containing 18% CO_2_ and 82% Ar with a flow rate of 15 l/min. Single-pass laser welding was performed in a He shielding gas environment using a TRUMPF TLF 6000 Turbo Laser CO_2_ with maximum power 6.5 kW.

During making the welded joints, different power parameters (voltage and current) and welding speeds were used, because the electrical heat input presented in units of Linear Welding Energy (LWE), *Q* (*Q* = *η*(*q*/*v*) (kJ/m or J/mm), where *q* (J/s)—welding torch arc power; *v* (mm/s)—weld torch advance rate, *η*—coefficient of beam absorption by the material, *η* = 0.9).

## 3. Hardness Distributions and Metallographic Tests

Hardness measurements and metallographic tests were carried out on welded joints made according to various technologies and regimes. Sections of welded joints were subjected to grinding, polishing, and etching with a 4% solution of HNO_3_ in C_2_H_5_OH (Nital). Then, the Vickers (HV) hardness measurements were made. The hardness was measured over the entire surface of the joint to covered zones BM, HAZ, and WM (Figure 2a). In joints made by conventional method, in which the size of the weld and HAZ zones were relatively large (Figure 2b,c), the distance between the impressions was equal to 1.0 mm and the measurement load 10.0 N. Hardness measurements on laser-made joints due to much smaller weld and HAZ dimensions were made with 0.5 mm increments and with a load of 5.0 N (Figure 2d). 

A three-dimensional representation of the hardness distributions determined on plane cross-sections for the respective welded joints is shown in Figure 3. For a welded joint made according to the Q1 regime, using a low-strength filler material (*R*_e_ = 470 MPa) and LWE = 1200 kJ/m, the zone with the lowest hardness level was in WM (200–220 HV) (Figure 3a). In the HAZ material at FL, the hardness increased to the level of 280-300 HV. The second area of low hardness occurred in the central part of HAZ (250–270 HV), and gradually increased to the level of 340–360 HV specific for BM.

In the joint made according to the *Q*2 regime (LWE = 1600 kJ/m), the lowest hardness values (270–280 HV) were recorded in the WM zone (Figure 3b). In the vicinity of FL, as in the joint material, the HAZ had a local maximum hardness (390–400 HV), while in the central part of the HAZ zone, the hardness again rapidly dropped (275–290 HV), and then increased to the BM level. 

When welding with high-strength filler material and LWE = 700 kJ/mm (*Q*3 regime), the minimum hardness values (260 ± 10 HV) were recorded in the HAZ zone (Figure 3c). In the WM, however, the hardness was high and reached 400 ± 10 HV.

When laser beam welding in accordance with the *Q*4 and *Q*5 regimes, the maximum hardness values of over 460 HV10 were recorded in the WM (Figure 3d). In the HAZ zone, the hardness decreased to a level some below that of the BM. However, as the LWE used during welding decreased, the hardness level in HAZ increased. Thus, when welding in accordance with the *Q*4 with LWE = 360 kJ/m, the minimum hardness level in the HAZ was 310 ± 10 HV; whereas, while welding according to the *Q*5 with LWE = 180 kJ/m, the minimum hardness level in the HAZ was 340 ± 10 HV, so it was comparable to that of BM.

In order to explain the changes of hardness levels in various welded joint zones, metallographic tests were performed using the JSM-7100F scanning electron microscope. The images in Figure 4, Figure 5 and Figure 6 show the microstructure of the welded material (a), and the microstructure of the HAZ material: In close proximity to the FL (b—HAZ1), in the centre of the HAZ (c—HAZ2), and at the end of the HAZ (d—HAZ3).

In the classic selection in HAZ, there are three areas distinguished: Coarse-grained (CGHAZ), intercritical (ICHAZ), and intercritically reheated coarse-grained (ICCGHAZ). However, this is a fairly arbitrary division, because depending on the welding conditions in these areas, a different type of microstructure of the material will arise. Also, for the laser welding proposed, we divided HAZ into four different sub-zones: Coarse-grained (CGHAZ), fine-grained (FGHAZ), intercritical (ICHAZ), and sub-critical (SCHAZ) adjacent to the BM [25]. In this article, the microstructure of HAZ material according to this division will be classified. 

In the case of the *Q*1 welding, a wide HAZ (5–6 mm) was observed in the joint. The WM has a finely grained ferritic microstructure (Figure 4a). This type of microstructure is created by using a lower strength of filler material and mainly as a result of large amounts of heat during welding according to the *Q*1 regime. WM obtained according to the *Q*1 welding regime has the lowest hardness in a welded joint (Figure 3a).

Microstructure type observed in the HAZ material directly at the FL is shown in Figure 4b. High-temperature heating over a relatively long period of time caused grains of recrystallization and the release of carbide particles and their coagulation. The hardness of the material in this HAZ area (CGHAZ) was slightly higher than in the WM (280–300 HV). In the central part of the HAZ (FGHAZ), the microstructure was made up of equiaxial fine, ferrite grains with a diameter of 1.0–5.0 μm (Figure 4c). Coagulated carbide particles are present in the grains, and residues of lamellar carbides are also found. The hardness of the material in this area is reduced to 250–270 HV (Figure 3a). In the microstructure of the area at the end of the HAZ (SCHAZ), we observed the initial stage of formation of equiaxial ferrite grains, with lamellar carbides visible in these grains (Figure 4d). Thus, it was a conglomerate that consisted of bainitic, ferritic, and perlite microstructures. The hardness of the material in this HAZ area was higher than in the central part and increased in the direction to the BM.

When welding according to the *Q*2 regime (filler material of a higher strength characteristics and a higher LWE level), a microstructure of bainite with a hardness of 290–310 HV was formed in the WM. An increase in hardness (380–400 HV) was observed in the HAZ at the FL, which resulted in the formation of a bainitic type microstructure with fine carbide precipitates with high dispersion (Figure 3b). In the central part of the HAZ and closer to the BM, microstructures of similar hardness and structure to that of *Q*1 regime were observed.

Microstructures for the respective areas of a welded joint according to the *Q*3 regime are presented in Figure 5. The use of a higher strength joint and a welding regime with a lower LWE in the WM resulted in a bainitic microstructure (Figure 5a), with a high level of hardness (390–410 HV) (Figure 3c).

In the HAZ material area directly at the FL, an increase in the number and coagulation of carbide precipitates was recorded in the initial microstructure as a result of heat exposure (Figure 5b). The microstructure hardness in this area was 350–380 HV and can be classified as mixture of bainite. In the central part of the HAZ, the microstructure was similar to that described in the *Q*1 regime. It was a mixture of fine equiaxial ferrite grains (1.0–5.0 μm), ferrite with coagulated carbide precipitates, and carbides in lamellar form (Figure 5c). In this area, the hardness was obtained at a minimum level for the joint. At the end of the HAZ, areas with equiaxial grains of ferrite and perlite were observed, while in others, the number of carbide precipitates from the microstructure of the base material increased (Figure 5d). The hardness level at the end of the HAZ increased and approached the hardness of the BM.

The microstructures formed in the joint during laser beam welding according to the *Q*5 regime are shown in Figure 6. During laser welding, the base metal was melted in the welding zone and then solidified. Since the width of the melted zone was small (~2.0 mm), cooling of the material occurred at a fast rate, and a microstructure of martensite and upper bainite with a hardness of 450–470 HV formed in the WM zone (Figure 3d and Figure 6a). A similar type of microstructure was also found in the HAZ material at the FL (Figure 6b). 

In the central part of the HAZ (~1.25 mm from the weld axis), the needle-like microstructure of the upper bainite transformed into a mixture of upper and lower bainite due to heat (Figure 6c). At a distance of ~2 mm from the weld axis (HAZ width ~1 mm), there was a thin band of a mixture of lower bainite and perlite microstructures (Figure 6d). The lowest hardness of the material in this area was achieved in laser weld joints. In the case of the *Q*5 regime, the hardness reduction was 330–340 HV, which was comparable to BM (Figure 3d). The increase in the LWE in welded process led to reduction in hardness of SCHAZ. For example, during welding in according to *Q*4 (LWE = 360 kJ/m)*,* a decrease in hardness to 300–310 HV was recorded in SCHAZ [30].

## 4. Strength Characteristics and Fracture Toughness the Materials of Welded Joints

### 4.1. Strength Characteristics and Fracture Toughness of BM 

The strength characteristics of base material–S960 steel were determined on flat rectangular specimens, 6 × 10 mm^2^, with 50 mm measuring section length, loaded according to uniaxial tensile test [31]. In order to determine the effect of temperature on the strength characteristics of S960 steel, tests were conducted in a temperature chamber. The environment of liquid nitrogen evaporation was used to obtain appropriate negative temperature. Cooling process control and temperature measurement were carried out by means of a controller, which allows to maintain the temperature level with ± 1 °C accuracy. 

Figure 7 shows of the nominal stress–strain curve graphs (*σ*–*e*) for different test temperatures. Lowering the test temperature leads to the increase of strength characteristics. Also, with the decrease in temperature, an increase in relative elongation of specimens was noted. A similar trend has also been observed for Hardox-400 high-strength steel [32]. 

Tests were also carried out to determine the characteristics of fracture toughness of the BM in the temperature range from −150 to 20 °C. The characteristics of fracture toughness were determined on specimens with the shape of a beam with a single edge notch during bending (SENB) with dimensions: *B* = 8.0 or 6.0; *W* = 16.0; *S* = 64.0 (mm) and *a*_0_/*W* ≈ 0.5. The fracture toughness was determined as a critical value of integral J, *J*_IC_ (or *J*_C_) based on ASTM Standard procedures [33]. Critical values of *J*_IC_ (or *J*_C_) were converted to fracture toughness units—*K*_JC_ according to the formula: *K*_JC_ = ((*E∙J*_IC_)/(1-*ν*^2^))^1/2^. The results of tests carried out to determine the characteristics of fracture toughness of the base material within the tested temperature range are presented in Figure 8. 

The image shows the load curves *P* - *u*_ext.,_ which are characterized by energy losses on the fracture process in the specimens (Figure 8a). In the temperature range from 20 to −20 °C, the development of a fracture occurred according to a ductile mechanism by the growth of voids, which are initiated around the particles of precipitates and inclusions (Figure 9a). The development of a fracture according to the ductile mechanism through the growth of voids required high levels of plastic strain and high energy, respectively, which translates into high values of fracture toughness, which were obtained during ductile cracking (Figure 8b). 

With a decrease in temperature, the levels of strength characteristics increased and the conditions necessary for the implementation of cleavage fracture occurred in the specimens—high levels of triaxiality ratio and normal stress components in the direction of the fracture are achieved [32,34]. In the transition area, a ductile fracture occurred directly in front of the tip of the pre-crack and a brittle fracture—at a certain distance from the tip of the crack (Figure 9b). This type of fracture development required less energy, which translates into lower critical values of fracture toughness.

In the temperature range *T* ≤ −100 °C, the load curves of the specimens were characterized by a straight line, which indicates the dominant share of elastic energy in the fracture development and, respectively, the occurrence of a completely brittle mechanism of cracking along the cleavage planes (Figure 9c). It should be stressed that high-strength ferritic steels with high levels of strength properties also exhibited high values of fracture toughness characteristics.

### 4.2. Strength Characteristics and Fracture Toughness of Welded Joints

The strength characteristics of the material of welded joints were determined on 0.5 × 2.0 mm flat specimens taken from different zones of welded joints. When cutting out the specimens, an electrical discharge machining method or a water jet was used to avoid introducing additional residual stresses (Figure 10a). Hardness was also measured on the side surfaces of the specimens, which allowed for additional material identification in welded joints position. Stress-strain curves and strength characteristics were obtained from uniaxial tensile tests [31]. The strength characteristics were determined as averages based on 3–5 tested specimens. Nominal values of strength characteristics for different zones of welded joints are presented in the Table 2. 

The fracture toughness characteristics were determined on SENB specimens with the dimensions, as for BM. In the SENB specimens, pre-cracks were introduced in different zones of welded joints (Figure 10b). Therefore, in order to identify the proper area, the etching of the specimen side surfaces and hardness measurements were carried out. As with BM specimens, fracture toughness was determined as a critical value of integral J, *J*_IC_ (or *J*_C_) based on ASTM Standard procedures [33]. Critical values of fracture toughness were determined as mean values based on testing at least three specimens, and are presented in the Table 2.

Based on the data obtained, it can be concluded that a lower yield strength, compared to the BM, occurred in the HAZ for all welding regimes. However, the difference between the yield strength of the BM and the HAZ material decreased with the reduction of the LWE level. When welding with a high level (LEW = 1200 kJ/m) of yield strength, values in the HAZ were equal to (0.55–0.60*) R*_e_ relative to the BM. When welding with a low level (LEW = 180 kJ/m), this reduction was less than 0.9*R*_e_ relative to the BM. This trend shows that reducing heat loss during welding has a positive effect on the level of strength characteristics in the HAZ.

The results obtained during the fracture toughness tests with pre-cracks introduced in various zones of welded joints allow us to conclude that low levels of the critical value of fracture toughness are found in areas of the material with hardness of about 400 HV. In laser beam welded joints, it is the area in the centre of the HAZ width; in conventional joints made according to *Q*3 regime, it is the WM. A bainitic microstructure occurred in these areas—a mixture of upper and lower bainite. The level of critical values of fracture toughness in these areas decreased to *K*_JC_ ≈ 140–160 MPa∙m^1/2^. However, this level was higher than the *K*_JC_ = 100 MPa∙m^1/2^, which characterizes the transition from ductile to brittle fractures [35]. For material with higher and lower hardness levels, higher values of fracture toughness were obtained. For a joint made in accordance to *Q*1 regime, the hardness level was in the 220–350 HV range and the fracture toughness was high, *K*_JC_ > 270 MPa∙m^1.2^.

The influence of negative test temperatures on the distribution of critical values of *K*_JC_ in the HAZ material for a welded joint according to the *Q*3 regime is presented in Figure 11. The change in fracture toughness *K*_JC_ in different zones of the joint at *T_test_* = −40 °C is shown in Figure 11a. The lowest fracture toughness characteristic of the material in the WM and the HAZ was at the FL. Towards the BM, increased *K*_JC_ values were observed. 

The change of *K*_JC_ value depending on temperature for the HAZ material at the FL and at the end of the HAZ is shown in Figure 11b. The difference in the value of *K*_JC_ during testing at *T*_test_ = −100 °C of the material in the entire HAZ is insignificant—from 75–96 MPa∙m^1/2^ for HAZ_FL to 95–111 MPa∙m^1/2^ for HAZ_NZ. At this temperature, cracking mainly occurred according to the cleavage mechanism, which, however, was preceded by a slight plasticization. As the temperature increased, the fracture toughness increased, as did the difference between the values of *K*_JC_ for the material at the FL and at the end of the HAZ.

## 5. Numerical Modeling the Welded Joints

Further tests were carried out to assess the state of stress and strain that occurs in different areas of the joint material of the welded components. Specimens containing welded joints were subjected to experimental uniaxial tensile tests. Force and elongation signals of the specimens were recorded under the loading. At the same time, while the specimens were subjected to loads, displacement fields were recorded on the surface of the specimens by means of the GOM digital video system Aramis. Using the ABAQUS software, numerical models of specimens containing welded joints were also created and subjected to simulation loads, identical to those on the specimens tested experimentally. Based on the data obtained, functional relations between hardness and material characteristics were established:*R*_e_ = 0.0008∙(*HV*)^2^ + 3.03∙(*HV*) − 145.03(1)
*R*_m_ = 0.00064∙(*HV*)^2^ + 2.81∙(*HV*) + 38.94(2)
*ε*_0_ = 27504500∙(*HV*)^−2.698^(3)

The value of Young’s modulus *E* was assumed as the average of the measurements made: *E* =204 GPa. The use of relations (1–3) allowed the material to be modelled for the respective zones of the welded joint even with the lack of tensile test results.

For modelling of the material from different areas of welded joints, the results of the uniaxial tensile tests presented in the previous chapter were used. Nominal strain and stresses were converted into true stress strain—ranging up to the beginning of the formation of the necking (*ε*_n_ ≤ *ε*_0_) according to the Formula:*ε*_t_ = *ln*(1 + *ε*_n_); *σ*_t_ = *σ*_n_(1 + *ε*_n_)(4)

In order to extend the relation *σ*_t_ = *f*(*ε*_t_), a linear approximation of the last ~200 points was applied (*σ*_ti_, *ε*_ti_) and the obtained relation was extrapolated on the range *ε*_t_ > *ε*_0t_, to the value *ε*_t_ = 3.0. The resulting relation *σ*_t_ = *f*(*ε*_t_) was additionally calibrated according to the procedures described in the papers [36,37,38]. 

Due to the symmetry of geometric shapes of the analyzed joints, one-half of the specimen containing the conventional welded joint and one-quarter of the joint made using laser beam welding were modelled. On the appropriate planes (YOZ), displacements in normal directions were blocked. The XOZ plane also blocked the possibility of displacement in the normal direction. This made it possible to simplify, reduce the number of finite elements, and at the same time, speed up calculations. 

The modelled welded joints were divided into zones, which corresponded, according to size, to the WM, HAZ, and BM for the analyzed joints (Figure 12). The HAZ was divided into three zones, HAZ1, HAZ2, and HAZ3. The size and shape of the respective zones in the welded joints were determined on the basis of microstructure observation by SEM and hardness measurements. The material properties in each modelled zone were entered in accordance with the procedures described above. Continuity of material was present between the different zones. Eight-node rectangular elements were used. The size of the elements was selected so as to ensure convergence of results and good compliance of the results from numerical calculations and measured by means of the DIC system on the surface of the specimens.

In numerical models, the load was generated by applying a value of displacement in the direction of loading to one end, while the other was blocked in place. The displacement level was applied in conformity with the displacement recorded during the uniaxial tensile test of the analyzed welded joints.

## 6. The Stress and Strain Analysis of the Welded Joints

Specimens containing welded joints, which were modelled as described in the previous section, were subjected to simulation loads identical to those used in the experiment. In order to verify the correctness of the results obtained, the numerically calculated strain fields on the outer surfaces of the specimens were compared with the results recorded using the GOM digital video system Aramis [39]. The figures present maps of strains counted numerically and obtained by means of a video system for selected specimens containing welded joints. 

Obtaining very similar strain distributions on the surface of welded joints calculated numerically and obtained by means of measurements with the use of the GOM video system proves the correct modelling of analyzed welded joints in terms of material selection to appropriate areas, their geometric dimensions, and the shape and dimensions of elements in the numerical program (Figure 13).

Numerical calculations of simulated loads allowed the assessment of the level of stress and strain inside the tested welded joints and using them as a basis allowed the prediction of the areas in which the damage will occur. The figures present maps of stresses (1) and strains (2) in the central (thickness-wise) plane of the specimens, which correspond to the maximum load carried by the joint, respectively, for (a)—*Q*1; (b)—*Q*3; (c)—*Q*5 (Figure 14).

According to the results of numerical calculations, the highest stress level was reached in the WM in joints made according to the *Q*1 regime and substantial strains occur in this area. This is because the yield strength of the material in a welded joint made according to the *Q*1 regime is ~535 MPa, and the maximum load results in a stress level above 600 MPa and a strain level above 20%. Since in other areas of a welded joint the yield strength of the material is higher, the joint will be damaged at the WM.

In the case of a joint made according to the *Q*3 regime, the highest levels of stress and strain were obtained in the HAZ material. At the highest load level of the joint, the stress value exceeded 900 MPa and the strain exceeded 10%. These values are higher than the material characteristics for the relevant zone—HAZ1. Therefore, damage of the welded joint was to be expected in the HAZ zone at the FL.

Stress and strain distributions for a laser welded joint made according to the *Q*5 regime indicate that the predicted area of damage is beyond the welded joint zone and the HAZ—in the BM. In the inner layers of the HAZ and the WM zones, the stress level did not exceed the yield strength of the materials, it is higher than the yield strength of the BM. 

The results obtained from the simulation of numerical loads are consistent with the results of experimental tests. During the uniaxial tensile testing of specimens, the development of the damage was recorded in those areas where the highest levels of strain were obtained during calculations. 

In Figure 15, stress and strain diagrams that correspond to the center line of the tested welded joints for the moment of reaching the maximum load are shown. Predicted destruction of elements may occur in areas where stress and strain peaks have been noted. For joints welded according to the *Q*1 regime, the maximum stress and strain that occurred in WM are shown in the results in Figure 14a and Figure 15a. The result of experimental test confirmed this prediction—the specimen was broken in weld material (Figure 16a). A similar situation was observed for the joint welded according to the *Q*2 regime (Figure 15b). 

In the case of a joint welded according to the *Q*3 regime, the maximum stress and strain were obtained in HAZ (Figure 14b and Figure 15c). Destruction of the specimen experimentally tested was also noted in this area (Figure 16b). In the joint welded according to the *Q*5 regime, the maximum stress and strain were obtained in BM (Figure 14c and Figure 15d). Experimental tests confirmed this prediction—the tested specimen broke in BM (Figure 16c).

## 7. Discussion and Conclusions 

As a result of the conducted research, extensive information was obtained regarding the strength of the base material and welded joints made of high-strength S960 steel. Thermo-mechanical processing during steel production leads to creation of the microstructure of tempered martensite—mixture of the upper and lower bainite. Because this steel is often used for the production of various types of elements and structures operated in conditions of Arctic aura, the results of tests in the range of reduced temperatures were presented. An interesting and important result is the increase in the strength (*R*_e_, *R*_m_) and plasticity (*ε*_0_, *A*_5_) characteristics with the temperature decreases (see Figure 8). It should be noted that a similar trend for those characteristics also noted for AHSS Hardox-400 [32]. This is a very advantageous feature that allows for great prospects in the use of AHSS under reduced temperatures.

The tested steel shows good fracture toughness (Figure 9). In the positive temperature range, the cracking process was carried out according to the ductile growth voids mechanism and the crack resistance was high (*K*_JC_ ≥ 240 MPa × m^1/2^), which is comparable to other ferritic steels. Along with the lowering of the temperature, the fracture toughness also decreased, but up to a temperature of −80 °C, the cracking mechanism was mixed–cracking occurred after high plasticization of the material, which led to fracture toughness at level *K*_JC_ ≥ 100 MPa × m^1/2^. Extensive research on the effect of temperature on the fracture toughness of specimens of various thicknesses is also provided in [40]. It should be emphasized that the tendency the fracture toughness of AHSS Hardox-400 is similar [32].

When welding elements, HAZ material of AHSS S960QC undergoes large changes due to the heat gradient coming from the weld. The impact of heat (in the work presented in LWE units) causes changes in the microstructure of the material, which results in a decrease in strength characteristics. It was found that the reduction of strength characteristics in the HAZ zone is proportional to the amount of heat consumed during welding. When welding with a high level of LWE (*Q*1, *Q*2, and *Q*3), the minimum level of yield strength in some areas of HAZ is 63%–78% compared to *R*_e_ for BM, and in other areas, HAZ is also lower than *R*_e_ for BM. The total width of the HAZ zone is 8–10 mm.

It has been shown that the use of laser welding, with a low level of LWE, slightly reduces the strength characteristics in HAZ. In welded joints, according to low-energy regimes (*Q*4 and *Q*5), the minimum values of strength characteristics are approximately 90%–95% from *R*_e_ of BM. The HAZ width in this welding regime is small, at 1.5–1.0 mm.

A method for assessing the strength of welded joints based on FEM numerical modeling and simulation of joint loads is presented. In order to model the material in individual joint areas, the results of experimental tests and the approximate dependencies of the material characteristics as a function of hardness were used (Formula 2). This approach allowed for the modeling of the material in all areas of the welded joints. The use of FEM made it possible to assess the stress and deformation levels in the inner layers of welded joints, and not only on the surface, as the methods using video systems allowed. The results presented in the article prove that the mechanical fields in the inner layers of welded joints significantly differ from those observed on the surface. The strength of the joints depends on the level of stress and deformation in the inner layers of the joints. As a result of FEM calculations, areas were determined in which the highest levels of plastic strains in welded joints took place, and which indicated the places of their potential destruction (Figure 15). 

In the *Q*1 (strength properties of filler material is lower and level of LWE is high) welded joint at the moment that corresponds to the highest load, according to FEM calculations, the highest level of deformations and stresses was obtained in WM (Figure 15a). During the experimental load, joint failure also occurred at this point at a tensile stress level of about ~ 570 MPa (Figure 14a), which corresponds to ~ 52% of the BM strength.

In the *Q*2 joint (high strength values of the filler material and high level of LWE), two areas of potential damage were determined—WM and HAZ (Figure 15b). Joint failure was experimentally recorded in WM at a tensile stress level of approximately 870 MPa, which corresponds to ~ 80% of the BM strength.

In the *Q*3 joint (high strength values of the filler material and middle level of LWE), HAZ was determined as the area of a potential damage (Figure 15c). Joint failure was experimentally noted in HAZ at a tensile stress level of approximately 850 MPa, which corresponds to 78% of the BM strength (Figure 14b).

In laser-welded joints, the LWE level used was lower: *Q*4 = 320 kJ/m and *Q*5 = 180 kJ/m, with smaller microstructure changes and reduction of strength properties in HAZ, respectively. Expected areas of destruction of welded joints were: For *Q*4, at the transition from HAZ to BM; and for *Q*5, at BM. During load of those welded joints, failure took place in BM at a tensile stress level of 1150–1180 MPa (Figure 14c and Figure 15d). Similar results of tensile strength of laser welded joints at comparable levels of LWE are presented in [25]. Based on the results presented, it can be concluded that welding with a low level of LWE does not lead to significant changes in the microstructure of HSS S960 steel, which in turn does not reduce the strength characteristics, and thus allows pne to obtain welded joints with a level of strength comparable to BM.

The lowest fracture toughness level of values (145–160 MPa × m^1/2^ at *T*_test_ = 20 °C) was observed in areas where the hardness is equal to 390–400 HW. At this level of critical values of fracture toughness, brittle fracture development is preceded by high plasticization of the material. However, as the temperature decreases, the plasticity part is decreased and at *T* < 100 °C, at the presence of a crack, it is expected that the material will be cracking fully brittle in these areas of the welded joints.

The main conclusions that can be express based on the presented in the paper research are as follows. 

During conventional welding, it is difficult (or impossible) to obtain a joint with comparable strength to BM. Even the use of a binder with a high level of strength and relatively low LWE in the welding process will cause the HAZ material to have an area with reduced strength characteristics, at the level of 75%–80% BM.Meanwhile, a laser welding with a low level of LWE allows you to make welded joints with a strength comparable to BM.

## Figures and Tables

**Figure 1 materials-13-00747-f001:**
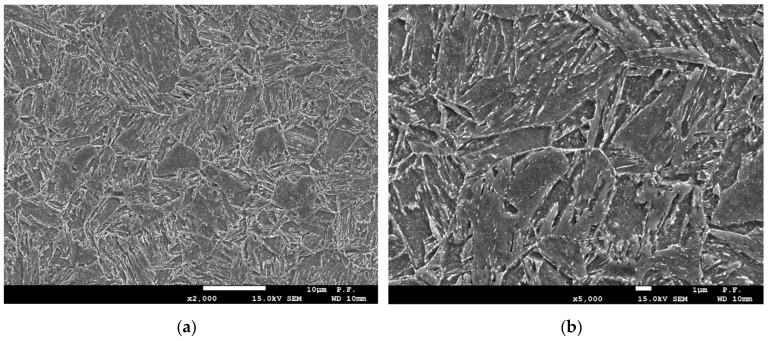
Microstructure of the base material, AHSS S960QC: (**a**)—2000×; (**b**)—5000×.

**Figure 2 materials-13-00747-f002:**
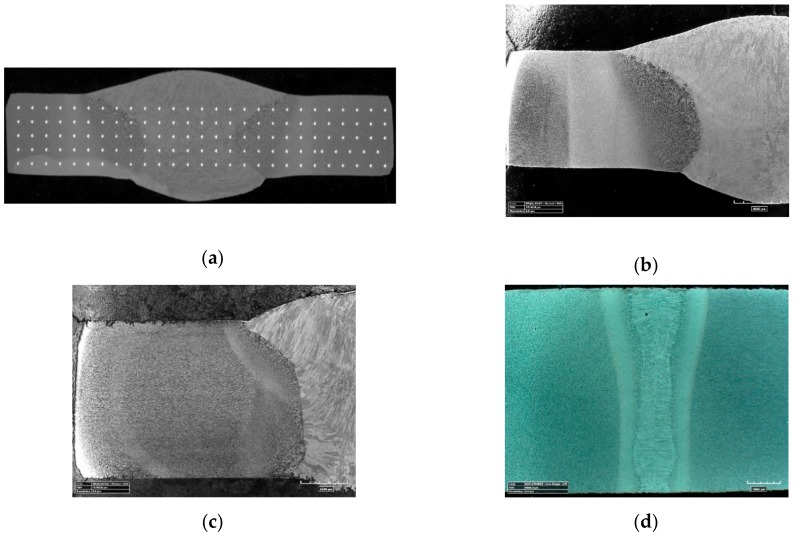
Views of cross-section of the welded joints with different level of Linear Welding Energy (LWE): (**a**) With measurement hardness points; (**b**) for *Q*1; (**c**) for *Q*3; (**d**) for *Q*5.

**Figure 3 materials-13-00747-f003:**
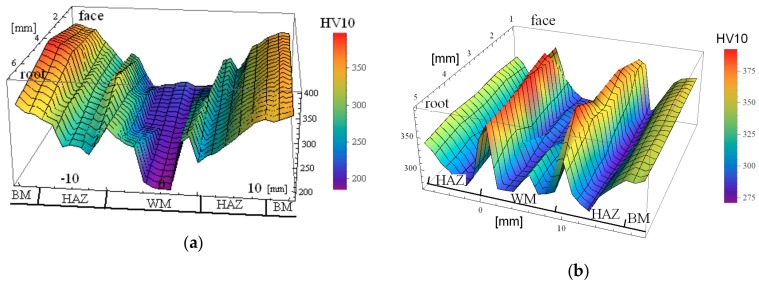
Hardness distribution of the analyzed welded joints: (**a**) *Q*1; (**b**) *Q*2; (**c**) *Q*3; (**d**) *Q*5.

**Figure 4 materials-13-00747-f004:**
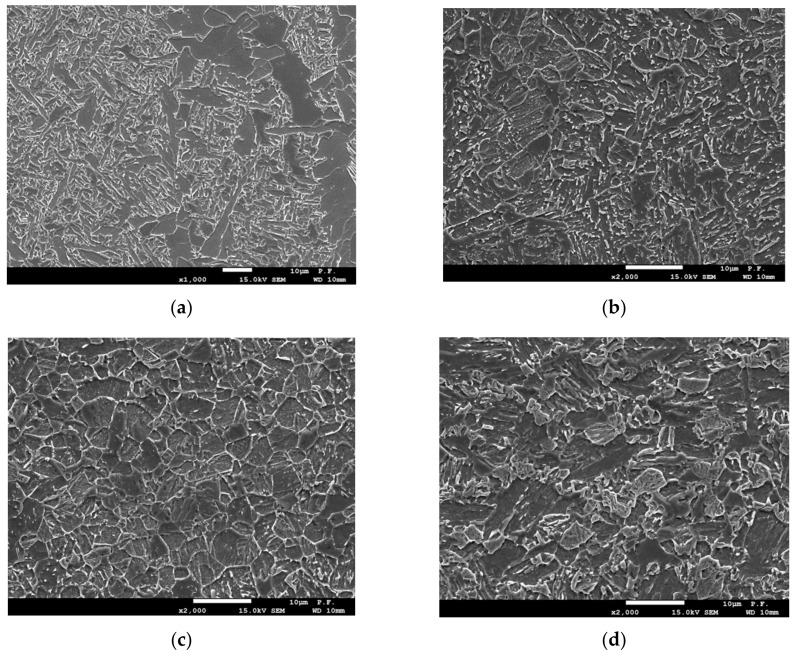
Microstructure in joint welded according to *Q*1: (**a**) WM; (**b**) HAZ1; (**c**) HAZ2; (**d**) HAZ3.

**Figure 5 materials-13-00747-f005:**
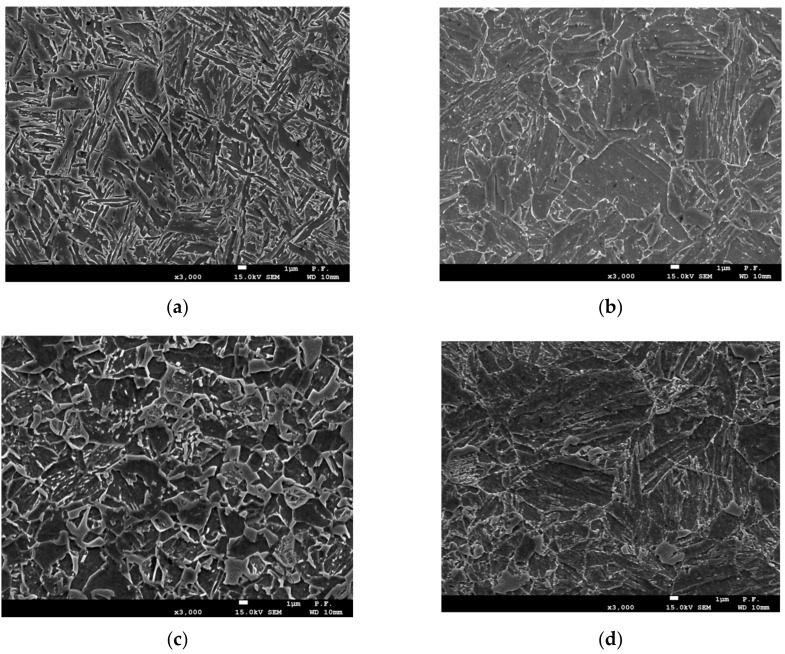
Microstructure in joint welded according to *Q*3: (**a**) WM; (**b**) HAZ1; (***c***) HAZ2; (**d**) HAZ3.

**Figure 6 materials-13-00747-f006:**
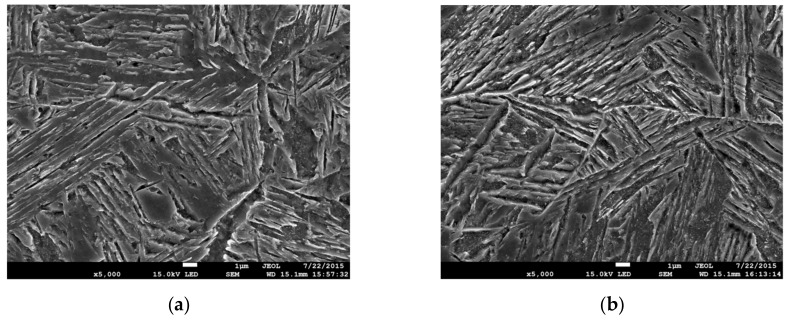
Microstructure in joint welded according to *Q*5: (**a**) WM; (**b**) HAZ1; (**c**) HAZ2; (**d**) HAZ3.

**Figure 7 materials-13-00747-f007:**
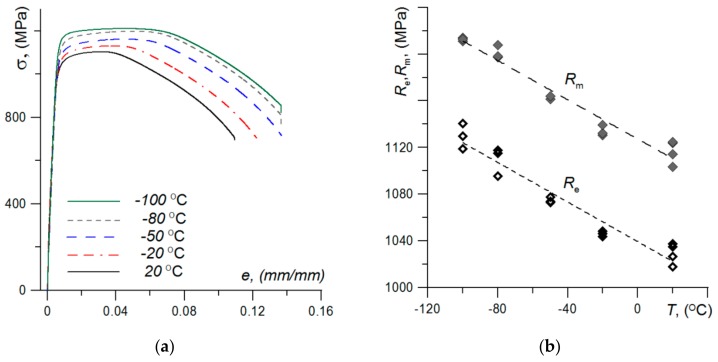
The tensile test results at reduced temperature: (**a**) diagrams of *σ*–*e*; (**b**) change of *R*_e_ and *R*_m_ values with temperature.

**Figure 8 materials-13-00747-f008:**
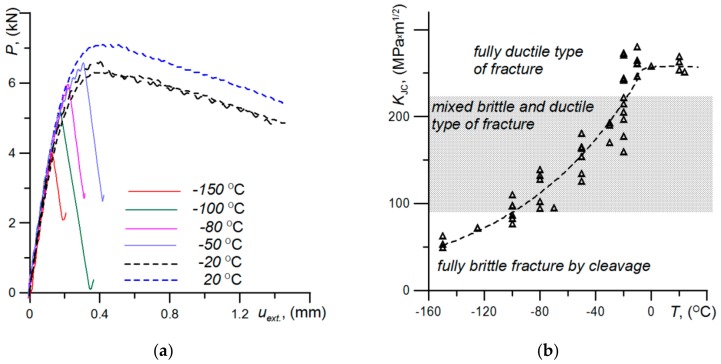
Fracture mechanisms of S960QC steel: (**a**) The diagrams of single edge notch during bending (SENB) specimens loading at different temperature; (**b**) three zones of different cracking types on brittle-to-ductile dependence.

**Figure 9 materials-13-00747-f009:**
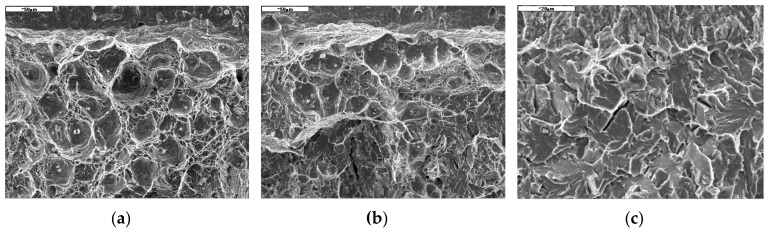
Types of cracking of AHSS S960QCat different temperatures: (**a**) *T* = 20 °C; (**b**) *T* = −50 °C; (**c**) *T* = −120 °C.

**Figure 10 materials-13-00747-f010:**
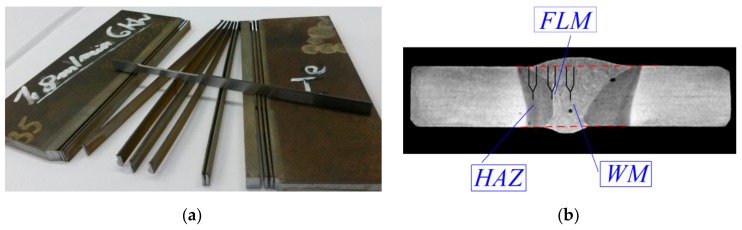
Making the specimens from different zones of welded joint to conduct tests: (**a**) Uniaxial tensile; (**b**) fracture toughness.

**Figure 11 materials-13-00747-f011:**
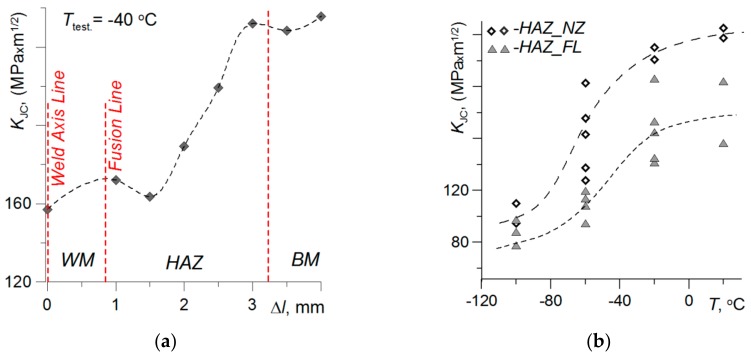
Critical values of *K*_JC_: (**a**) In deferent zones of welded joint; (**b**) temperature influence on change of *K*_JC_ in material HAZ-FL and HAZ-NZ.

**Figure 12 materials-13-00747-f012:**
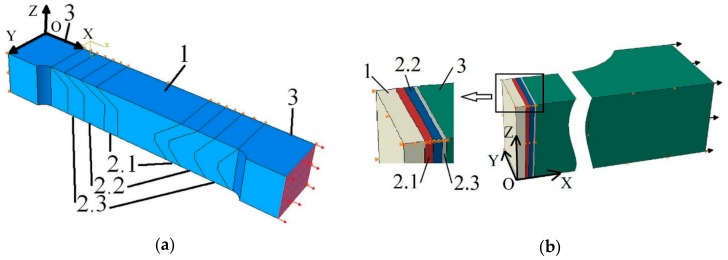
Numerical models of the welded joints: (**a**) For conventional type; (**b**) for laser type (marking in figure: WM – 1; HAZ arias—2.1, 2.2, 2.3; BM—3).

**Figure 13 materials-13-00747-f013:**
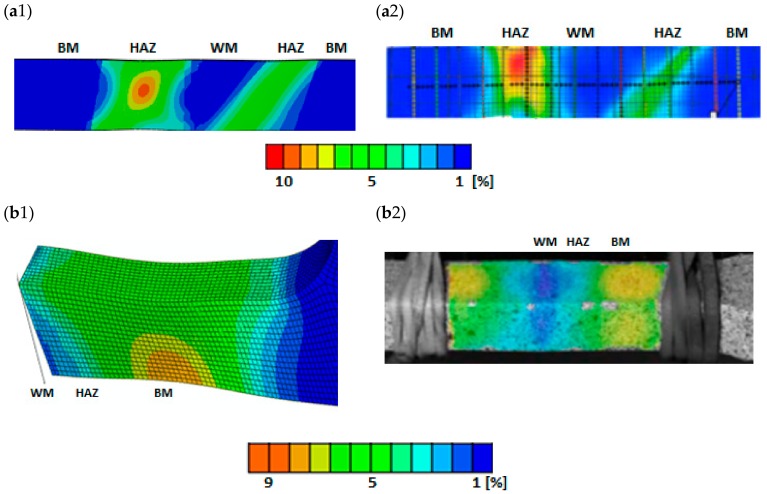
The fields of strains obtained according to FEM (**a**1; **b**1) and recorded using video-system (***a***2; **b**2) for welded joints *Q*3 (**a**) i *Q*5 (***b***).

**Figure 14 materials-13-00747-f014:**
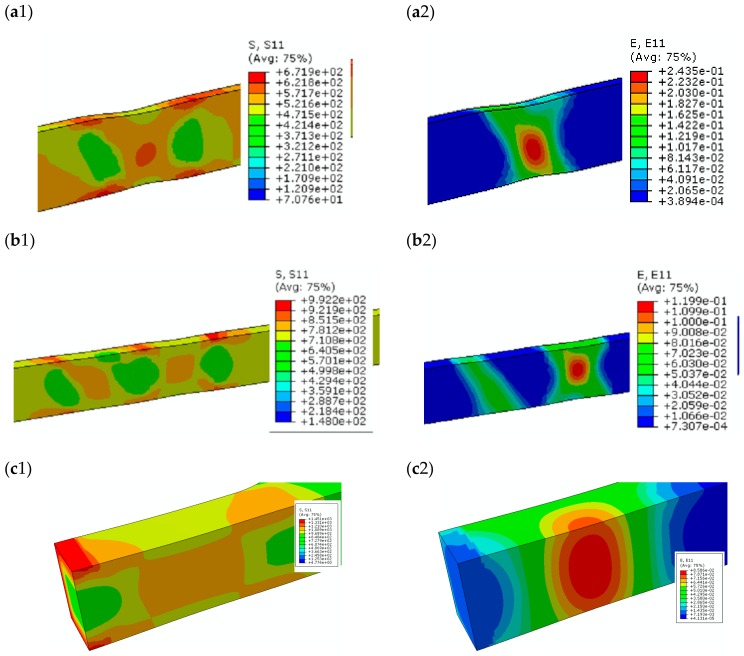
The fields of stresses (1) and strains (2) in middle plane of the welded joints made according to regimes: (**a**) *Q*1; (**b**) *Q*3; (**c**) *Q*5.

**Figure 15 materials-13-00747-f015:**
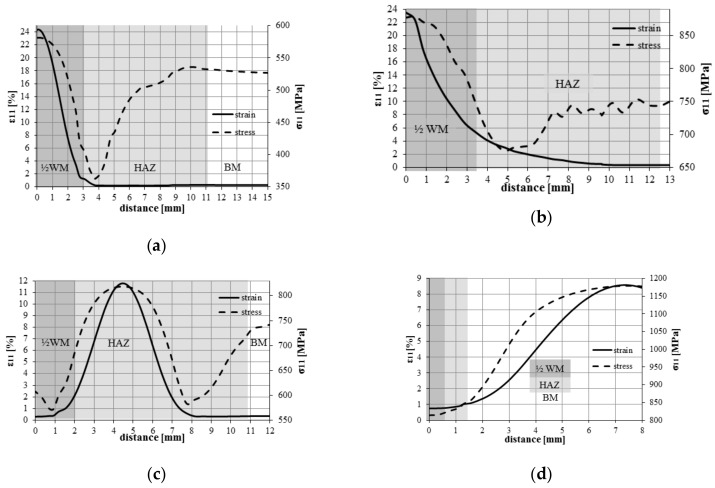
Stress and strain distribution along the center line inside the welded joints: (**a**) *Q*1; (**b**) *Q*2; (**c**) *Q*3; (**d**) *Q*5.

**Figure 16 materials-13-00747-f016:**
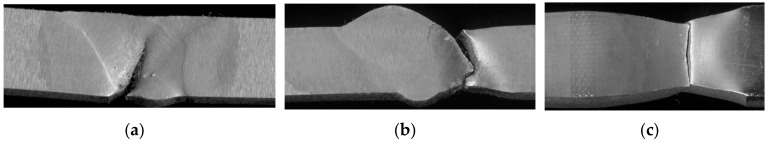
Views of broken welded joints made according to the regimes: (**a**) *Q*1 in WM; (**b**) *Q*3 in HAZ; (**c**) *Q*5 in BM.

**Table 1 materials-13-00747-t001:** Parameters of welding processes of S960 steel.

	*B* (mm)		Filler Material	*R*_e_ (MPa)	*R*_m_ (MPa)	*A*_5_ (%)	Protective Environment	*LWE*
								*Q* (kJ/m)
*Q*1	8.0	MAG	OK12.51	470	560	26	M21, 15 l/min	1200
*Q*2	6.0	MAG	Union X90	890	950	15	M21, 15 l/min	1600
*Q*3	8.0	MAG	Union X96	930	980	14	M21, 15 l/min	700
*Q*4	6.0	Laser	without	–	–	–	He, 12 l/min	320
*Q*5	6.0	Laser	without	–	–	–	He, 12 l/min	180

**Table 2 materials-13-00747-t002:** Mechanical properties of materials from different zones of the welded joints.

	Q, (kJ/m)	Q*, (J/mm^2^)	Weld Zone	HV10	E, (GPa)	R_e_, (MPa)	R_m_, (MPa)	ε_0_, (%)	K_JC_, (MPam^1.2^)
1	1200	150	WM	226 ± 12	196	475 ± 13	570 ± 15	12.2 ± 1.5	287 ± 06
			HAZ1	250 ± 07	220	595±11	747 ± 12	6.1 ± 1.2	303 ± 08
			HAZ2	285 ± 10	206	677 ± 07	867 ± 11	4.1 ± 0.7	281±03
			HAZ3	346 ± 15	212	1009 ± 10	1127 ± 08	3.9 ± 0.3	274±05
2	1600	267	WM	310 ± 10	204	750 ± 12	860 ± 10	4.1 ± 0.5	185± 09
			HAZ1	381 ± 12	196	978 ± 11	1043 ± 10	2.9 ± 0.2	237 ± 05
			HAZ2	282 ± 15	201	748 ± 07	863 ± 11	4.3 ± 0.7	191 ± 03
			HAZ3	345 ± 13	207	870 ± 08	1015 ± 10	3.9 ± 0.4	265 ± 11
3	700	87	WM	393 ± 10	178	1050 ± 08	1090 ± 12	3.1 ± 1.3	157 ± 08
			HAZ1	296 ± 06	204	778 ± 12	816 ± 14	6.4 ± 0.7	172 ± 09
			HAZ2	349 ± 11	185	936 ± 09	995 ± 08	3.6 ± 0.3	165 ± 06
			HAZ3	371 ± 12	191	1083 ± 11	1138 ± 11	2.9 ± 0.2	277 ± 05
4	320	40	WM	457 ± 12	198	1153 ± 12	1225 ± 12	1.6 ± 0.3	165 ± 05
			HAZ1	470 ± 11	–	–	–	–	–
			HAZ2	395 ± 15	207	1033±10	1087 ± 11	2.6 ± 0.4	150 ± 11
			HAZ3	320 ± 10	204	810 ± 11	915 ± 10	3.7 ± 0.4	275 ± 09
5	180	23	WM	450 ± 15	193	1057 ± 13	1355 ± 12	1.7 ± 0.3	167 ± 07
			HAZ1	465 ± 10	–	–	–	–	–
			HAZ2	405 ± 09	192	1104 ± 12	1426 ± 14	2.5 ± 0.3	145 ± 05
			HAZ3	343 ± 06	204	930 ± 11	1040 ± 07	4.1 ± 0.4	180 ± 08
	BM		BM	350 ± 12	187	1010 ± 08	1100 ± 10	3.9 ± 0.4	253 ± 11

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
