# Peer review of "Influence of LWE on Strength of Welded Joints of HSS S960—Experimental and Numerical Analysis"

_materials, 2020, doi:10.3390/ma13030747_

Round 1

Reviewer 1 Report

Dear Authors,

thank you for your contribution to materials. You present an analysis on the strength of weld joins by experimental analysis and FEM calculations. Your contribution is well-prepared, strong in content and scientifically sound. However, I have found two minor issues:

l.101: „A three-dimensional representation of the hardness distributions for the respective joints is
102 shown in Figures 3.“

the distribution if three-dimensional, since it is measured in a plane. This is still the case, if you create a 3D surface plot. Please correct.

l. 245: please indicate in the table caption the meaning of orange table cells

Author Response

Dear Authors,

thank you for your contribution to materials. You present an analysis on the strength of weld joins by experimental analysis and FEM calculations. Your contribution is well-prepared, strong in content and scientifically sound. However, I have found two minor issues:

l.101: „A three-dimensional representation of the hardness distributions for the respective joints is
102 shown in Figures 3.“

The distribution if three-dimensional, since it is measured in a plane. This is still the case, if you create a 3D surface plot. Please correct.

245: please indicate in the table caption the meaning of orange table cells.

The lowest value of yield strength and fracture toughness for the respective welded joints were marked orange . In present version orange marking in table 2 removed.

Thank You very much for the high rating of our work.

Reviewer 2 Report

Dear authors! Your manuscript looks very attractive for readers and science community at present form.

Good luck!

Author Response

Dear authors! Your manuscript looks very attractive for readers and science community at present form.

Good luck!

Thank You very much for the high rating of our work.

Reviewer 3 Report

Scale bars on figure 2 is not clear.

There are grammatical mistakes throughout the manuscript. Lines 386 and 387 are repeated.

Please describe the assumptions used in the finite element simulation. How does the assumptions affect the welding characteristics?

Please provide an explanation for the graph 15c.

It is recommended to rearrange the conclusion section while highlighting the significant findings of the study.

Author Response

Thank you for reviewing our paper and for its positive assessment.

The English language has been improved.

Scale bars on figure 2 is not clear.

The thickness of the tested plates was 6 or 8 mm.

There are grammatical mistakes throughout the manuscript. Lines 386 and 387 are repeated.

The duplicate sentence in line 387 has been deleted. (it was a technical error)

Please describe the assumptions used in the finite element simulation.

Additional information for the description of the numerical models has been introduced.

How does the assumptions affect the welding characteristics?

I do not understand this question. When it comes to calculated stress distributions in welded joints, they have insignificantly changed depending on the size of the elements used.

Please provide an explanation for the graph 15c.  

Additional explanations of the  numerical calculations results, including those shown in Fig. 15c, has been provided in the text.

It is recommended to rearrange the conclusion section while highlighting the significant findings of the study.

The last section has been corrected.

Round 2

Reviewer 3 Report

The authors have addressed all the major concerns.